# Sodium, Potassium and Iodine Intake in an Adult Population of Lithuania

**DOI:** 10.3390/nu14183817

**Published:** 2022-09-16

**Authors:** Urte Zakauskiene, Ernesta Macioniene, Lina Zabuliene, Diana Sukackiene, Ausra Linkeviciute-Dumce, Valdas Banys, Nomeda Bratcikoviene, Dovile Karosiene, Virginija Slekiene, Virginijus Kontrimas, Kazys Simanauskas, Algirdas Utkus, Deimante Brazdziunaite, Vilma Migline, Indre Makarskiene, Ingrida Zurlyte, Ivo Rakovac, Joao Breda, Francesco P. Cappuccio, Marius Miglinas

**Affiliations:** 1Institute of Clinical Medicine, Faculty of Medicine, Vilnius University, LT-03101 Vilnius, Lithuania; 2Centre of Nephrology, Vilnius University Hospital Santaros Klinikos, LT-08661 Vilnius, Lithuania; 3Institute of Biomedical Sciences, Faculty of Medicine, Vilnius University, LT-03101 Vilnius, Lithuania; 4Faculty of Fundamental Sciences, Vilnius Gediminas Technical University, LT-10223 Vilnius, Lithuania; 5Family Medical Center Pilenai, LT-36243 Panevezys, Lithuania; 6Outpatient Clinic Nefrida, LT-93220 Klaipeda, Lithuania; 7Community Well-Being Center, Mykolas Romeris University, LT-08303 Vilnius, Lithuania; 8Ministry of Health, LT-01402 Vilnius, Lithuania; 9WHO Country Office, LT-05011 Vilnius, Lithuania; 10WHO European Office for the Prevention and Control of Noncommunicable Diseases (NCD Office), Division of Country Health Programmes, WHO Regional Office for Europe, 2100 Copenhagen, Denmark; 11WHO Athens Quality of Care Office, 10675 Athens, Greece; 12World Health Organization Collaborating Centre for Nutrition, Warwick Medical School, University of Warwick, Coventry CV4 7AL, UK; 13Department of Medicine, University Hospital Coventry and Warwickshire NHS Trust, Coventry CV2 2DX, UK

**Keywords:** Lithuania, salt, sodium, potassium, iodine, population, 24 h urine collection

## Abstract

Hypertension is a leading risk factor for cardiovascular events and death. A reduction in salt intake is among the most cost-effective strategies to reduce blood pressure and the risk of cardiovascular diseases. Increasing potassium lowers blood pressure and is associated with lower cardiovascular risk. Adequate iodine intake is important to prevent iodine deficiency disorders. Salt iodization is a key strategy to prevent such deficiency. In Lithuania, no surveys have been performed to directly assess sodium, potassium and iodine consumption. The aim of the present study was to measure sodium, potassium and iodine intake in a randomly selected adult Lithuanian adult population using 24 h urine collections, and to assess knowledge, attitudes and behavior towards salt consumption. Salt and potassium intakes were estimated in 888 randomly selected participants by 24 h urine sodium and potassium excretion and 679 individuals provided suitable 24 h urine samples for the analysis of iodine excretion. Average salt intake was 10.0 (SD 5.3) g/24 h and average potassium intake was 3.3 (SD 1.3) g/24 h. Only 12.5% of participants consumed less than 5 g/24 h of salt. The median value of urinary iodine concentration (UIC) was 95.5 μg/L. Our study showed that average salt intake is twice as high as the maximum level recommended by the World Health Organization while potassium and iodine intakes in Lithuania are below the recommended levels.

## 1. Introduction

Non-communicable diseases (NCDs) are the leading causes of death and disability globally and their reduction by modulating modifiable risk factors is a health priority [1] and has substantial economic and social benefits. Unhealthy diet and excessive sodium intake increase the risk of high blood pressure (hypertension), a leading risk factor for cardiovascular disease (CVD) [2]. Hypertension affects more than 1.13 billion people worldwide and is among the leading causes of serious cardiovascular events and premature death [3,4]. Over half of middle-aged adults in Lithuania have hypertension [5]. Due to the growing burden of CVD worldwide and in view of low implementation costs, a population reduction in salt intake is among the most cost-effective strategies to tackle the epidemic of hypertension and CVD. Reduced salt consumption will improve overall population health, extend life expectancy and decrease mortality [6].

The World Health Organization (WHO) recommends a universal reduction in sodium intake to less than 2000 mg (or less than 5 g of salt) per day [7]. However, in many countries, the average daily intake of sodium still remains 4000 mg (10 g of salt) or more [6,8,9]. Population-based prospective studies show that each daily increment of 1000 mg in sodium excretion is associated with an 18% increase in cardiovascular risk [10].

Therefore, in the Global Action Plan for the Prevention and Control of NCDs, the WHO suggests a 30% reduction in salt intake [1]. An assessment of the population salt consumption is essential for the planning and monitoring of the effectiveness of national salt reduction plans [11].

The WHO also recommends an increase in potassium intake from food to reduce the risk of CVDs in adults, such as hypertension, coronary heart disease and stroke [12,13,14]. The higher sodium-to-potassium ratio in 24 h urine collections is a stronger predictor of the risk of CVD than either nutrient status alone [15,16].

Moreover, since 1994, the WHO and the United Nations International Children’s Emergency Fund (UNICEF) reached an agreement to recommend Universal Salt Iodization as a key strategy for the prevention and elimination of deficiency disorders (IDDs) worldwide. The recommended daily iodine intake for adults of 150 μg/24 h is sufficient to meet physiological requirements and to achieve an adequate intake of the nutrient [17]. According to the WHO, iodine deficiency is the single greatest cause of preventable mental retardation in children [17,18]. Dietary iodine deficiency in adults causes thyroid disorders, such as goiter, increased risk of thyroid cancer and hypothyroidism [19,20]. On the other hand, excessive intake of iodine should be avoided, since it can result in thyroid function disorders, especially in previously iodine-deficient areas [21].

Since 2013, the WHO has initiated joint standardized assessments of population salt and iodine status to unify the assessment and monitoring of sodium and iodine intake and to develop public health policies [1]. As more than 90% of ingested sodium and iodine appears in the urine in the following 24 to 48 h, urinary sodium and iodine excretions directly reflect daily dietary intake of sodium and iodine [22,23]. Currently, twenty-four-hour (24 h) urine collections are the recommended method to monitor population sodium and potassium intake and to investigate the associations between sodium and potassium intake and health outcomes [24]. The impact of salt iodization on population iodine intake is monitored using a few biomarkers of iodine nutrition: urinary iodine concentration (UIC), thyroglobulin concentration in blood and goiter rate [25]. The average 24 h urinary iodine excretion is considered the best indicator of the habitual iodine intake of a given population [26]. Therefore, 24 h urine collection is the main investigative method for sodium, potassium and iodine intake status.

The aim of the present study was to investigate for the first time sodium, potassium and iodine intake level in a randomly selected adult population using 24 h urine collection and to study knowledge, attitudes and behavior towards salt consumption in Lithuania.

## 2. Materials and Methods

The study administration group and the WHO Regional Office in Europe organized and held a training workshop for project group members in January 2019. This study was carried out between September 2019 and November 2020, as part of the project “Assessment of sodium and iodine status in the Lithuanian population and development of public health policy guidelines” supported by Vilnius University hospital Santaros klinikos, Vilnius University, in collaboration with the State Public Health Promotion Fund and the WHO. One thousand and thirty-four men and women, aged 18–69 years, were examined. Ethical approval was obtained from the Vilnius Region Bioethics Committee. The survey was carried out in accordance with the Declaration of Helsinki and Good Clinical Practice. All participants provided written informed consent to take part in this study.

### 2.1. Sample Size 

To obtain a nationally representative sample, the country was divided into three regions—North, South-East, and West—and proportional sampling was applied to reflect sex, age and residents of each of the three regions (Figure 1). Population data were provided by the Department of Statistics of Lithuania.

### 2.2. Sample Selection and Invitation Procedure

Random sampling was used to obtain a list of potential participants in the survey. According to PAHO/WHO, EMRO-WHO and WHO/EURO Protocols [27,28,29], people were excluded if younger than 18 or older than 69 years; with a known history of heart, kidney or liver disease, stroke; were prescribed therapy with diuretics (in the last two weeks prior to the urine sample collection date); terminally or mentally ill; and pregnant women. Participants were invited during an appointment with their family physician or by a phone call. During the first visit, those who were unable to give an informed consent and those with any other condition that would make 24 h urine collection impossible were excluded.

After signed informed consent, every participant was attributed a bar code to allow anonymization. They were handed a self-reported questionnaire, detailed written instructions on how to correctly collect 24 h urine, and two 2.5 L plastic polyethylene containers for the collection. The questionnaire and the containers were marked with the barcode assigned to the subject.

Participants completed an anonymous questionnaire, designed specifically for this study, to determine knowledge, attitudes and behavior towards salt consumption. Finally, they returned both the questionnaire and the 24 h urine collection to a personal health care facility. Detailed information about the inclusion and exclusion procedures is presented in Figure 2. Out of the initial sampling frame, 43.4% (1082/2492) agreed to take part and consented. An additional 47 participants did not provide urine collections. Quality control applied to the returned collections excluded a further 147 participants; 103 participants had inappropriate 24 h urine collections (1 reporting > 1 missing void, 3 with total urine volume < 500 mL and 99 with either under-collections (<20 h) or over-collections (>26 h) or unknown duration); finally, 44 participants had total creatinine excretion outside 2 standard deviations of the sex-specific distribution of urinary creatinine in the sample. A final sample of 888 participants (82.1% of those participating) with complete urine collections was included in the final analysis. For iodine status analysis, 209 subjects with thyroid disorders or known consumption of iodine supplements were also excluded.

### 2.3. Questionnaire and Physical Data

The questionnaire consisted of 39 questions covering demographic and social status (6 questions), anthropometry (2 questions), health status (5 questions), salt consumption and dietary characteristics (25 questions), and physical activity (1 question) (see Appendix A). Height, weight, blood pressure (BP) and heart rate were measured in all participants with validated equipment and standardized protocols, as also described elsewhere [27,28,30,31]. Hypertension was defined as systolic and/or diastolic BP ≥ 140/90 mmHg or being on current antihypertensive treatment [27].

### 2.4. Procedures of 24 h Urine Collection

Each participant received clear verbal explanation and detailed written instructions on how to collect 24 h urine. Participants were instructed to collect their urine into 2.5 L plastic polyethylene containers for 24 h, starting with an empty bladder. They discarded the first void in the morning, and collected all the urine until the following morning, including the first void. Investigators asked participants if they forgot or spilled urine samples. Urine collections deemed to be incomplete 24 h urine samples were excluded from this study (Figure 2).

### 2.5. The 24 h Urine Sample Preparation in the Laboratory

Once in the laboratory, total 24 h urine volume was recorded and three 12 mL aliquots were taken from each sample. All urine samples were frozen and stored at −20 °C until the chemical analysis. One aliquot was used for the determination of sodium, potassium, and creatinine. This sample was thawed and centrifuged with 1.500 RCF in ambient temperature for 10 min before analysis. The second aliquot was used to determine iodine concentration and was thawed just before the analysis, mixed well to ensure the homogeneity of the sample. The third aliquot was left as back-up, if needed.

### 2.6. Assurance of the 24 h Urine Sample Quality

We applied a robust process of quality assurance to avoid inclusion of incomplete urine collections (either under- or over-collected) [28]. Urine were excluded from the analysis if: The start and end day and time of the collection were not recorded and could not be ascertained;The duration of the collection, either observed by staff or self-reported, was out of range of 20–26 h;If self-report of missing more than one void using a questionnaire;If 24 h urine volume was less than 500 mL;If 24 h urinary creatinine excretion was outside 2 standard deviations of the sex-specific distribution (5.9–26.0 mmol/24 h for men and 4.0–16.4 mmol/24 h for women).

### 2.7. Urine Biochemistry Panel

Biochemical parameters (sodium, potassium and creatinine) in urine were analyzed on an Abbott Architect ci8200 analyzer with the dedicated reagents from the same manufacturer (Abbott, Chicago, IL, USA) (Table 1). UIC was determined using the Sandell-Kolthoff method which has been validated by analyzing the EQUIP (Ensuring the Quality of Urine Iodine Procedures) robust external quality assurance program samples provided by CDC (Centers for Disease Control and Prevention, Atlanta, GA, USA). Study samples digestion step was performed in standard dry block heaters (VWR International bvba, Leuven, Belgium). The colorimetric measurements were performed in an Agilent 8453 UV- spectrophotometer (Agilent Technologies, Shanghai, China). Potassium iodate (KIO_3_) was used as calibrator (Sigma-Aldrich, St. Louis, MO, USA). Ammonium peroxydisulfate ((NH_4_)_2_S_2_O_8_), arsenic trioxide (As_2_O_3_), ceric ammonium sulfate dihydrate ((NH_4_)_4_Ce (SO_4_)_4_ 2H_2_O) and sodium chloride (NaCl) were of analytical grade (Sigma-Aldrich Chemie GmbH, Steinheim, Germany). Concentrated sulfuric acid (H_2_SO_4_) was from EMD Millipore (Darmstadt, Germany). Deionized water was at least of 18 MΩ∙cm resistivity and was produced by Millipore Direct Q water purification system (Merck KGaA, Darmstadt, Germany).

### 2.8. Statistical Analysis

We used graphical and descriptive data analysis for an estimation of the measures of central tendency and variability in the sample and in groups; χ^2^ test for an identification of differences in proportions, *t*-test and its non-parametric analogue—the Wilcoxon signed-rank test, analysis of variance (ANOVA) and the non-parametric Kruskal–Wallis test, for an identification of differences in groups, multiple linear and logistic regression for an estimation of associations and strength of dependence between variables. Different methods were applied according to the aim and purpose of the analysis and satisfaction of parametric test assumptions. Statistical analysis and data handling were carried out using R Project for Statistical Computing.

The urinary excretion of sodium (NaU in mmol/24 h), potassium (KU in mmol/24 h) and iodine (in mcg/L) were recalculated to a daily intake. Estimations were made using approximate equalities: 1 mmol = 22.99 mg of sodium, 1 mmol = 39.1 mg of potassium. The conversion from dietary sodium (Na) intake to salt (NaCl) intake was made by multiplying the sodium value by 2.542. Then, sodium values were multiplied by 1.05 (assuming that approximately 95% of sodium ingested is excreted in the urine) [22,23]. Potassium dietary intake was calculated assuming 85% of the potassium ingested is excreted in the urine [32]. Urinary iodine was expressed in mcg/24 h using multiplier 1.08 assuming 92% of iodine ingested is excreted in the urine. The cut-off values according to the WHO/EURO protocol were used [28]: <5 g/24 h for salt consumption, >90 mmol/24 h for potassium consumption, for iodine consumption: insufficient (<100 μg/L) with subcategories: severe (<20 μg/L), moderate (20–50 μg/L), mild (50–100 μg/L); adequate consumption [100–200 μg/L); consumption above requirement (200–300 μg/L); excessive consumption (≥300 μg/L).

## 3. Results

### 3.1. Baseline Characteristics of the Participants 

A final population sample of 888 participants (mean age 47.4 years; 47.5% men) were included in the final analysis (Figure 1). Baseline characteristics are shown in Table 2. Women were older (48.6 vs. 46.1 years), had a lower average BMI (25.9 vs. 27.2 kg/m^2^) and a lower blood pressure (systolic 119.1 mmHg vs. 129.7 mmHg, diastolic 75.7 mmHg vs. 80.4 mmHg); 35.5% of participants were hypertensive (systolic BP ≥ 140 mmHg and/or diastolic BP ≥ 90 mmHg or on regular antihypertensive treatment) and 17% were current smokers. The prevalence of hypertension (42.5% vs. 29.2%, *p* < 0.001) and smoking (22.0% vs. 12.4%, *p* < 0.001) was higher in men than in women.

### 3.2. Urinary Sodium Excretion and Salt Consumption 

Table 3 shows the results in detail. Mean urinary volume was slightly higher in men than women. Men had higher creatinine excretion than women (16 mmol/24 h vs. 10.2 mmol/24 h, *p* < 0.001). Mean urinary sodium excretion was 162.4 mmol/24 h and calculated mean salt intake was 10.0 g/24 h. Men had significantly higher sodium excretion (191.3 mmol/24 h vs. 136.3 mmol/24 h, *p* < 0.001).

Gender, body weight, potassium excretion and systolic blood pressure were significantly associated with 24 h sodium excretion (Table 4).

A total of 12.5% of participants fulfilled the WHO recommendations on daily salt intake: only 4.3% of men and 20% of women consumed < 5 g of salt per day (*p* < 0.001). The majority of participants (87.5%) consumed > 5 g of salt per day: 47.3% of participants had 5–10 g salt intake per day, 26% of participants consumed 10–15 g salt per day and 14.2% of participants consumed > 15 g salt per day (Figure 3).

### 3.3. Urinary Potassium Excretion and Potassium Intake

Mean urinary potassium excretion was 73.8 mmol/24 h and the calculated mean of potassium intake was 3.3 g/24 h (Table 3). Men had significantly higher potassium excretion than women (80.1 mmol/24 h vs. 68.1 mmol/24 h, *p* < 0.001); therefore, they consumed more potassium than women (3.6 g/24 h vs. 3.1/24 h, *p* < 0.001); 23.1% of participants fulfilled the WHO recommendations on potassium intake per day, more often men than women (30.8% vs. 16.1%, *p* < 0.001). The mean sodium-to-potassium ratio was 2.3, with a spread across the population examined (Figure 4). It was significantly higher in men (2.6 vs. 2.1, *p* < 0.001) compared to women (Table 3).

### 3.4. Daily Intake of Iodine and Use of Iodized Salt 

A total of 679 participants (average age of men was 46.5 years; women 48.1 years) were included into the final analysis of iodine status. Our study showed that in Lithuania iodine intake is insufficient. The median value of urinary iodine concentration (UIC) was 95.5 μg/L.

With reference to the WHO criteria for the assessment of the population iodine status, urinary iodine excretion was adequate in 32.8% of participants (Table 5). Iodine consumption was above requirement in 10.3% or excessive in 4.6% of the participants, irrespective of sex or area of residence. Only 54.9% of participants indicated that they use iodized salt for salting food at home.

## 4. Discussion

This is the first national study of sodium, potassium and iodine consumption in Lithuanian adults, using the gold-standard measure of 24 h urinary sodium, potassium and iodine excretion as biomarkers of intake. The results show unequivocally that salt consumption is high, potassium consumption is lower than recommended, both in men and in women, whilst iodine intake is still inadequate in more than half the sample studied.

### 4.1. Salt Consumption

Average population salt consumption was 10.0 g/24 h, double the WHO recommended maximum population target of 5 g/24 h. Only one in ten participants met these targets. Importantly, one-third of the participants (40.2%) showed a very high salt consumption (more than 10 g/24 h). Men had significantly higher salt consumption than women (11.7 g/24 h vs. 8.4 g/24 h). In addition to sex, body weight, potassium excretion and systolic blood pressure were significantly related to 24 h sodium excretion. Hypertension and overweight were also more common amongst men. These gender differences are not unexpected. Men usually have a higher food intake, which translates into a higher salt intake. In addition, Lithuanian men live unhealthy lifestyles. Our questionnaire revealed that women are less likely to eat salty processed meat, frozen and processed fish. Compared to women, a larger proportion of men consume crisps, roasted nuts, smoked, salted or cured meat products, prepared soups and stock cubes for infusion and drink natural mineral water. In addition, men use these products more often than women and eat food cooked outside the home more often. Moreover, the majority of men do not think about regulating their salt consumption.

On average, Eastern European countries consume excessive amounts of salt (10.8 g/24 h in Moldova, 11.6 g/24 h in Montenegro, 12.6 g/24 h in Ukraine) [30,31,33]. Salt intake is very high in Asian countries (18.8 g/24 h in Kazakhstan) and North China (15.6 g/24 h) [34,35]. Among Nordic European populations, a study from Finland showed a salt intake of 10.5 g in men and 8.2 g in women [36]. The population-based Tromsø Study investigated salt intake in Norwegian women and men. Mean sodium excretion was 4.09 ± 1.60 and 2.98 ± 1.09 g/24 h in men and women, respectively, corresponding to a calculated salt intake of 10.4 and 7.6 g [37]. A Swedish study of randomly selected young men found that the mean excretion of sodium was 198 mmol, corresponding to 11.5 g NaCl intake [38]. Powles and colleagues in their systematic review found that estimated mean sodium intake in Western Europe ranges from 3.28 to 4.43 g/24 h, equivalent to a salt intake of 8.33 to 11.25 g/24 h [39]. Powles et al. also reported that in 5 studies from Eastern and Central Europe the estimated average sodium intake varied from 3.92 to 4.41 g/24 h corresponding to a salt intake of 9.8 to 11 g/24 h. Our results for Lithuanian adults are settled in the middle of this range.

The link between increased salt intake and high blood pressure leading to cardiovascular disease, stroke and kidney disease is well known [40]. An independent association between sodium intake and cardiovascular disease risk has been described in the Nordic cohorts in Europe. Higher sodium intake of 100 mmol was linked to 51% higher coronary heart disease mortality, 45% higher cardiovascular disease mortality, and 26% higher all-cause mortality [41]. A recent meta-analysis looked at the relationship between salt intake, as determined by repeated measurements of 24 h urinary sodium excretion, and cardiovascular disease in 10,709 healthy adults. The hazard ratio (HR) for those in the highest quartile of sodium excretion was 1.60 compared with the HR for those in the lowest quartile. This meta-analysis showed that every 1000 mg increase in sodium excretion per day (2.54 g in salt) was associated with an 18% increased risk of cardiovascular disease [10]. New research shows that dietary salt is associated with a higher risk of premature death and shorter life expectancy, regardless of diet, lifestyle, socio-economic status and pre-existing conditions. In a study of more than 500,000 people, those who always added salt to their diet were 28% more likely to die prematurely (before the age of 75) than those who never or rarely added salt [42].

In our population-based study, salt intake greatly exceeded the recommended levels. This reaffirms the urgent need for concerted action to address salt consumption in Lithuania. This includes a setting up a working group to develop strategies to reduce salt levels in foods, such as food reformulation, front-of-pack labeling, consumer education, establishing monitoring systems to track salt intake, implementing legislative policies and environmental interventions [11]. Regular salt intake should be part of an active approach to cardiovascular disease prevention and control.

### 4.2. Potassium Consumption

Average population excretion of potassium was 73.8 mmol/24 h (equivalent to approximately 3.3 g/24 h), lower than the WHO recommended minimum level of 90 mmol/24 h, equivalent to approximately 3.90 g/24 h. Only approximately one in five participants in our study met these WHO targets. A meta-analysis by Ma et al. showed that each daily increment of 1 g of potassium excretion was associated with an 18% decrease in cardiovascular risk [10]. A recent study has shown that replacing conventional (sodium) salt with potassium-enriched salt significantly reduced the incidence of cardiovascular disease and deaths in a large high-risk Chinese population [43].

A large body of data shows that a high Na/K ratio is of great importance for high blood pressure and cardiovascular events. A long-term study of a cohort in Japan showed that dietary Na/K ratio is a major risk factor for stroke, cardiovascular disease and all-cause mortality [44]. The urinary sodium-to-potassium molar (Na/K) ratio in our study averaged 2.3, significantly higher in men than in women (2.6 vs. 2.1). In comparison, in the Norwegian Tromsø Study the Na/K ratio was approximately 1.8 in both genders [37]. Na/K ratio measured in our study was also substantially higher than that reported for men and women in Finland (2.08 and 1.92) [36]. The International Collaborative Study on Salt (INTERSALT) study conducted in the late 1980s estimated the mean molar Na/K of Western populations to be 2.98. The INTERSALT study also showed that lowering the population average urinary Na/K from 3.1 to 1.0 would lower the population blood pressure by 3 to 5 mmHg, resulting in an 8–14% reduction in stroke mortality, 5–9% reduction in CHD mortality and 4–7% reduction in all-cause mortality [32]. Thus, measuring the ratio is important: a combination of low sodium intake and high potassium intake may be a better predictor of cardiovascular disease risk than sodium and potassium intake alone. According to the WHO, the optimal Na/K ratio when following recommended intakes is approximately one [7].

### 4.3. Iodine Consumption

This is the first nationally representative adult population-based study in Lithuania to assess iodine status. In Lithuania, the only population study of iodine status was carried out in children in 1995. In 2087 children aged 8–10 years spot urine analysis revealed a median UIC of 75 μg/L, with mild to moderate iodine deficiency in some areas [45]. In 2005 Lithuania adopted a policy of universal mandatory salt iodization and other measures for the control of iodine deficiency disorders. Our study showed that after 17 years iodine intake in Lithuania has increased but it still remains insufficient (current median of UIC was 95 μg/L). In comparison with other European countries, where iodine status has been studied in adult populations, the situation in Lithuania is more favorable than in Italy [46], where median UIC was 46 (23–88) µg/L [46], and comparable to that seen in Belgium [47], that recorded a median UIC of 93.6 µg/L [47].

Our study provides further evidence to support the fact that 52.3% of Lithuanian adults still had insufficient intake of iodine and severe iodine deficiency was detected in 1 of 10 study participants. Less than 5% of participants had excessive iodine intake (UIC above 300 μg/L). Similar or even worse situation was seen in Italy. Over 70% of the Italian study population had an inadequate iodine intake in both women and men, but iodine inadequacy was more severe in women [46]. However, only 28.6% of the adult population in Moldova in 2016 had insufficient iodine consumption (equally distributed by sex or area of residence), and 2.3% had severe iodine deficiency [30]. According to an analysis of 40 studies, Ittermann et al. showed that iodine deficiency is still present in Europe and in 7 out of 13 (53.8%) studies of adults the median standardized UIC was less than 100 μg/L and the cause of persistent iodine inadequacy was most likely attributable to an insufficient use of iodized salt [48].

According to the questionnaire survey, 54.9% of our study participants used iodized salt. Another survey of eating habits of 2573 adults in Lithuania in 2019 demonstrated that 57.2% of respondents used iodized salt at home [49]. Similarly, Spanish researchers have found that in 2011–2012 in Madrid, iodized salt was commonly consumed by 59% participants [50]. However, a study in Belgium in 2009 found that iodized salt was consumed at home by only 36.8%. respondents [51]. 28% (2.6 g/24 h) of the median salt intake of the German population (9.3 g/24 h) was calculated as iodized salt, compared to 43% in Switzerland [52].

Since salt is an iodine-rich carrier, it is important to reconcile efforts to reduce salt with efforts to maintain adequate iodine intake. This requires a concerted effort to monitor changes in intake and adjust the amount of iodine added to the salt consumed [53].

### 4.4. Strengths and Limitations

The strengths of our study were: (a) the large representative random sample of the population (men and women) that covered the entire Lithuanian territory; (b) the use of the gold-standard method of 24 h urine collection to measure sodium, potassium and iodine excretion, indicators of intake; (c) the use of careful instructions on how to collect complete urine samples and strict quality control to minimize under- and over-collections; (d) the application of external iodine quality control; (e) the exclusion of participants on thyroid medications and iodine supplements.

Our study also had limitations: (a) the possibility of a self-selection bias due to participation rate remains a possibility; (b) although the information about the use of iodized salt was collected, exact iodine concentration in the salt remains unknown; (c) we did not have biochemical information on thyroid status (such as thyroid hormones, thyroglobulin, and thyroid volume).

## 5. Conclusions

In Lithuania, salt consumption is unequivocally high, with a large share of the population with very high salt intake, potassium consumption lower than recommended, both in men and in women, and inadequate iodine intake in more than half population studied. The present study conducted in the period 2019–2020 on a large national sample of the general adult population represents an important reference for future monitoring of sodium, potassium and iodine intake in Lithuania. It will serve for the development of comprehensive policies that would allow the country to address in tandem the challenges of very high salt and inadequate intakes of potassium and iodine. A balanced approach is needed and may be obtained by promoting the availability of less salt with higher iodine content [53].

## Figures and Tables

**Figure 1 nutrients-14-03817-f001:**
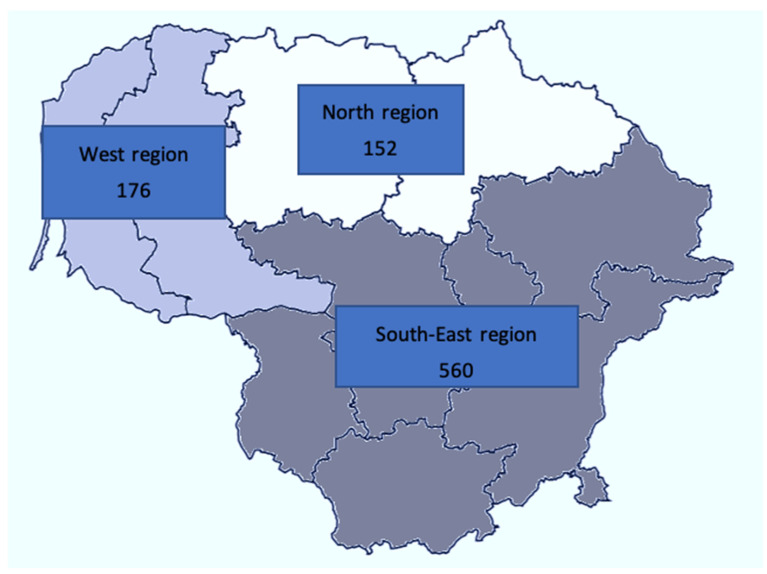
Number of participants in each region.

**Figure 2 nutrients-14-03817-f002:**
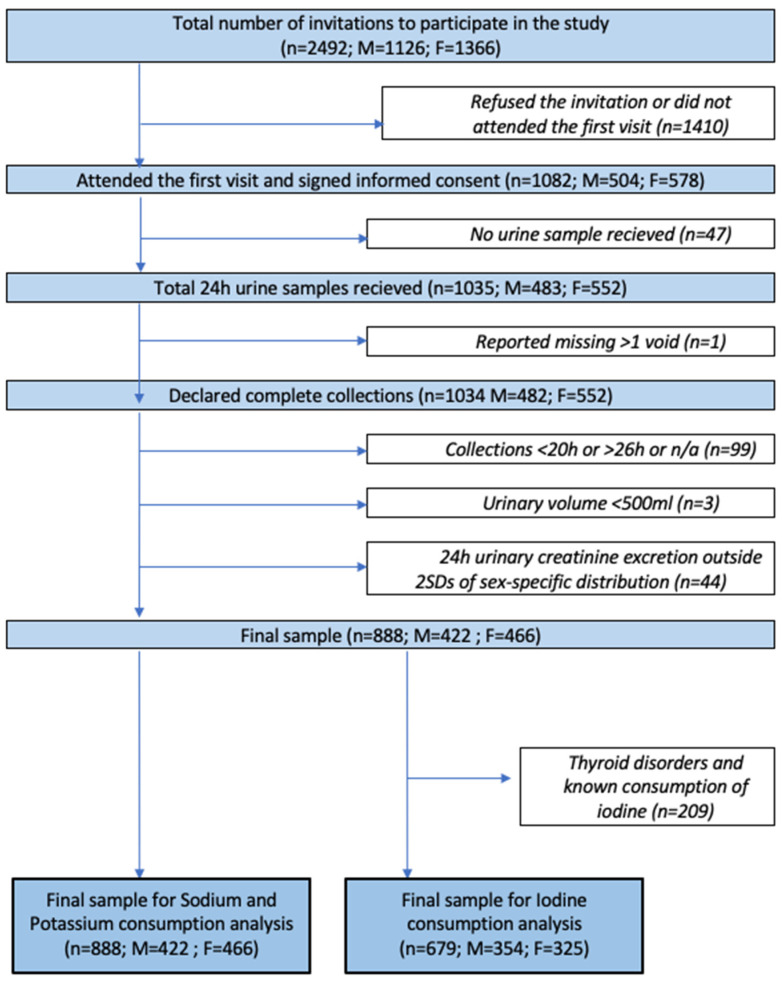
Procedure for participant inclusion and exclusion to this study.

**Figure 3 nutrients-14-03817-f003:**
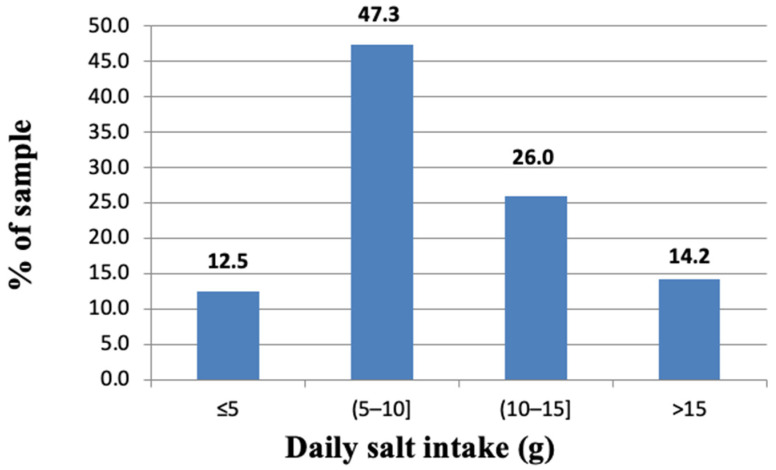
Distribution of single 24 h salt intake estimates.

**Figure 4 nutrients-14-03817-f004:**
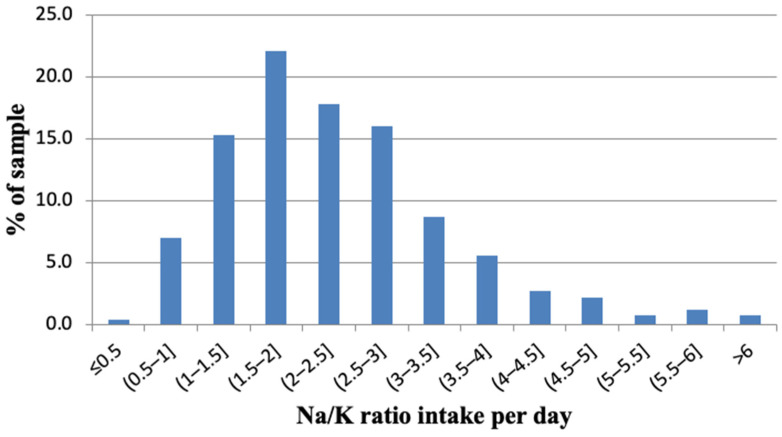
Distribution of the sodium-to-potassium ratio.

**Table 1 nutrients-14-03817-t001:** Summary of methods of urine biochemistry panel and their traceability information.

Analyte (Urine)	Method	Traceability to Reference Material
Creatinine	Photometric, enzymatic	NIST SRM 967
Sodium, potassium	Indirect potentiometry (indirect ion-selective electrode (ISE))	NIST SRM 918 and NIST SRM 919
Iodine	Spectrophotometric, Sandell–Kolthoff method	NIST RM2670a, SRM3668

**Table 2 nutrients-14-03817-t002:** Baseline characteristics of the participants.

Variables	Total	Men	Women	*p*-Value
	888	422	466	-
Age, year	47.4 (12.1)	46.1 (12.3)	48.6 (11.7)	0.82
Height, cm	173.2 (9.4)	180.4 (6.8)	166.8 (6.2)	<0.001
Weight, kg	79.7 (16.7)	88.5 (14.8)	71.9 (14.4)	<0.001
BMI, kg/m^2^	26.5 (4.7)	27.2 (4.2)	25.9 (5)	<0.001
Systolic BP, mmHg	124.1 (22.5)	129.7 (28.1)	119.1 (14.1)	<0.001
Diastolic BP, mmHg	77.9 (9.3)	80.4 (8.9)	75.7 (10.1)	<0.001
Pulse rate, bpm	72.5 (10.0)	72.3 (10,1)	72.8 (9.9)	0.55
Hypertension, *n* (%)	312 (35.5)	177 (42.5)	135 (29.2)	<0.001 *
Current smokers, *n* (%)	151 (17.0)	93 (22.0)	58 (12.4)	<0.001 *

Results are the mean (SD) or as percentage. BMI—body mass index; BP—blood pressure. *p*-value indicates statistical difference between men and woman. * *p*-value indicates the significance of the difference in proportion.

**Table 3 nutrients-14-03817-t003:** The 24 h urine analysis: volume, excretions of sodium, potassium, creatinine; estimated salt and potassium intake.

Variables	Total	Men	Women	*p*-Value
*n*	888	422	466	
Volume, ml/24 h	2045 (799) 1920 (1472; 2530)	2057 (823) 1920 (1487; 2559)	2035 (778) 1920 (1470; 2553)	0.68
Sodium, mmol/24 h	162.4 (86) 146.1 (104.8; 198.6)	191.3 (94.6) 171.2 (131.8; 236.3)	136.3 (67.5) 123.7 (89.7; 167.4)	<0.001
Potassium, mmol/24 h	73.8 (29.6) 69.9 (52.9; 88.4)	80.1 (32.7) 75 (56.9; 98.1)	68.1 (25.2) 65.6 (49; 82.6)	<0.001
Sodium-to-potassium ratio	2.3 (1.1) 2.1 (1.6; 2.9)	2.6 (1.2) 2.4 (1.8; 3.2)	2.1 (0.9) 1.9 (1.4; 2.7)	<0.001
Creatinine, mmol/24 h	12.9 (4.5) 12.3 (9.5; 15.8)	16 (4.1) 15.9 (13.3; 18.7)	10.2 (2.6) 10.1 (8.3; 11.9)	<0.001
Salt intake, g/24 h	10 (5.3) 9 (6.4; 12.2)	11.7 (5.8) 10.5 (8.1; 14.5)	8.4 (4.1) 7.6 (5.5; 10.3)	<0.001
Potassium intake, g/24 h	3.3 (1.3) 3.1 (2.4; 4)	3.6 (1.5) 3.4 (2.6; 4.4)	3.1 (1.1) 2.9 (2.2; 3.7)	<0.001
Salt < 5 g/24 h, (*n*%)	111 (12.5)	18 (4.3)	93 (20)	<0.001 *
Potassium > 90 mmol/24 h, (*n*%)	205 (23.1)	130 (30.8)	75 (16.1)	<0.001 *

Results are the mean (SD) and the median (25; 75th percentile) or *n* (%). *p*-value indicates statistical difference between men and woman. * *p*-value indicates the significance of the difference in proportion.

**Table 4 nutrients-14-03817-t004:** Variables associated with 24 h urinary sodium excretion.

Variables	Coefficient	95% CI	*p*-Value
Association with 24 h sodium excretion (mmol/24 h)			
Male sex	21.04123	(9.88; 32.21)	<0.01
Body weight (kg)	1.132687	(0.78; 1.49)	<0.01
Potassium excretion	1.103706	(0.94; 1.27)	<0.01
Systolic blood pressure	0.389656	(0.04; 0.74)	0.029

*p*-value indicates statistical significance in linear regression.

**Table 5 nutrients-14-03817-t005:** Distribution of participants’ iodine status according to the WHO criteria based on 24 h urinary iodine concentrations (μg/L).

Iodine Status (Iodine Concentrations, μg/L)	N (%)
All participants	679
Insufficient (<100):	355 (52.3)
Severe insufficiency (<20)	75 (11)
Moderate insufficiency (20; 50)	88 (13)
Mild insufficiency (50; 100)	192 (28.3)
Adequate consumption (100; 200)	223 (32.8)
Above requirement (200; 300)	70 (10.3)
Excessive consumption (≥300)	31 (4.6)

Results are the number (%).

## Data Availability

Data available on request in writing to the corresponding author.

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
