# Peer review of "Sodium, Potassium and Iodine Intake in an Adult Population of Lithuania"

_nutrients, 2022, doi:10.3390/nu14183817_

Round 1

Reviewer 1 Report

Thank you for conducting this key study and presenting the results so clearly! Below I list a few minor editorial changes. Please change your text to read as follows:

+ line 47: "... Lithuania have arterial"

line 69: "...adults of 150 ug/day is sufficient"

line 127: " ...marked with the barcode"

line 128: "Participants completed"

line 130: "... along with the 24-hour urine sampled collected in the disposable"

line 141: "...thyroid disorders or known consumption"

line 156: "... for 24 hours as comfortable"

line 200: " A set of statistical..."

line 238: Please move Table 3 until after Table 3 is mentioned in the text.

line 256: Please move Figure 3 to after Table 4 so the figures and tables follow the text order. 

line 365: "... status study in the children was conducted..."

line 371: "...deficiency disorders, iodine..."

line 374:  "... in Italy [46] and comparable to the seen in Belgium [47]."

line 412: Please move the subhead " Our study also had limitations to the left margin and remove the bullet so that it matches the "strengths" subhead.

416: " remains unknown; and 

Author Response

Dear Reviewers,

The authors are grateful for the reviews and opportunity to improve our article “Sodium, Potassium and Iodine Intake in an Adult Population of Lithuania”. We revised the article and corrected grammatical and spelling mistakes. The manuscript was edited by native English-speaking colleague who contributed to the article as one of the authors.  Your comments and remarks were truely valuable. Please find response to comments below.

Response to Reviewer 1 comments

Reviewer 1:Thank you for conducting this key study and presenting the results so clearly! Below I list a few minor editorial changes. Please change your text to read as follows:

Comment 1: line 47: "... Lithuania have arterial"

Response: Thank You for your comment. We made changes to the manuscript.

Comment 2: line 69: "...adults of 150 ug/day is sufficient"

Response: Thank You for your comment. We made changes to the manuscript.

Comment 3: line 127: " ...marked with the barcode"

Response: Thank You for your comment. We made changes to the manuscript.

Comment 4: line 128: “Participants completed”

Response: Thank You for your comment. We made changes to the manuscript.

Comment 5: line 130: "... along with the 24-hour urine sampled collected in the disposable"

Response: Thank You for your comment. We made changes to the manuscript.

Comment 6: line 141: "...thyroid disorders or known consumption"

Response: Thank You for your comment. We made changes to the manuscript.

Comment 7: line 156: "... for 24 hours as comfortable"

Response: Thank You for your comment. We made changes to the manuscript.

Comment 8: line 200: " A set of statistical..."

Response: Thank You for your comment. We made changes to the manuscript.

Comment 9: line 238: Please move Table 3 until after Table 3 is mentioned in the text.

Response: Thank You for your comment. We mentioned Table 3 earlier in the text.

Comment 10: line 256: Please move Figure 3 to after Table 4 so the figures and tables follow the text order. 

Response: Thank You for your comment. We move Figure 3 after Table 4.

Comment 11: line 365: "... status study in the children was conducted..."

Response: Thank You for your comment. We made changes to the manuscript.

Comment 12: line 371: "...deficiency disorders, iodine..."

Response: Thank You for your comment. We made changes to the manuscript.

Comment 13: line 374:  "... in Italy [46] and comparable to the seen in Belgium [47]."

Response: Thank You for your comment. We made changes to the manuscript.

Comment 14: line 412: Please move the subhead " Our study also had limitations to the left margin and remove the bullet so that it matches the "strengths" subhead.

Response: Thank You for your comment. We removed bullets.

Comment 15: line 416: " remains unknown; and”.

Response: Thank You for your comment. We made changes to the manuscript.

Reviewer 2 Report

The manuscript by Urte Zakauskiene et al. focuses on examining the sodium, potassium, and iodine intake levels of Lithuanian adults using 24-hour urine collection. The manuscript is well-organized. However, the manuscript should be revised for the English language and style, and some comments are suggested for publication:

1. The abstract is missing a sentence discussing why measuring potassium is important.

2. The abstract (Page 1, lines 30-32) mentions that 1034 participants were examined by assessing salt, iodine, and potassium intake by 24-hour urine sodium, iodine, and potassium excretion. However, only 888 participants were examined for sodium and potassium consumption analysis, and 679 participants for iodine consumption analysis. This should be corrected.

3. The acronym UNICEF on Page 2 line 66 is incorrect. This should be corrected.

4. Figure 1 on Page 3 should be fixed. The numbers for the West and North regions are cut.

5. A supplemental document can be provided including the questionnaire given to the participants mentioned on Page 5 lines 144-153.

6. Is the sentence on Page 5 line 163 another sub-section?

7. The Materials and Methods (Page 5 line 167) mentions that chloride, albumin, and urea were determined in the urine. These results can be included in the manuscript to determine if participants suffer from kidney function and/or improper fluid cell balance.

8. The percentage mentioned on Page 6 line 226 is for women. This should be corrected.

9. Is the measurement g/d or mmol/d the same as g/24h or mmol/24h? The manuscript should be consistent.

10. As a suggestion, Table 4 should be placed before Figure 3 on Page 8. In the manuscript Table 4 is mentioned first than Figure 3.

11. The p-value on Page 9 line 264 should be p<0.001.

12. The title of Iodine status should be µg/l, it's missing information.

13. The percentage on Page 10 line 295 should be 40.2%. In Figure 3 the number of participants that consumed 10-15 g of salt per day was 26% and 14.2% for participants that consumed > 15 g of salt per day.

14. Is the sentence on Page 12 line 412 another sub-section?

15. The manuscript should focus on and mention that men have significantly different results leading to poorer outcomes than women. Where there any significant responses in the questionnaire that can answer why? Example: Demographic and Social status, Health status, or Lifestyle habits (physical activity).

Author Response

Dear Reviewers,

The authors are grateful for the reviews and opportunity to improve our article “Sodium, Potassium and Iodine Intake in an Adult Population of Lithuania”. We revised the article and corrected grammatical and spelling mistakes. The manuscript was edited by native English-speaking colleague who contributed to the article as one of the authors.  Your comments and remarks were truely valuable. Please find response to comments below.

Response to Reviewer 2 comments

Reviewer 2: The manuscript by Urte Zakauskiene et al. focuses on examining the sodium, potassium, and iodine intake levels of Lithuanian adults using 24-hour urine collection. The manuscript is well-organized. However, the manuscript should be revised for the English language and style, and some comments are suggested for publication:

Comment 1. The abstract is missing a sentence discussing why measuring potassium is important.

Response: Thank You for your comment. We added missing information about potassium to the abstract: “Increasing potassium lowers blood pressure and is associated with lower cardiovascular risk”.

Comment 2. The abstract (Page 1, lines 30-32) mentions that 1034 participants were examined by assessing salt, iodine, and potassium intake by 24-hour urine sodium, iodine, and potassium excretion. However, only 888 participants were examined for sodium and potassium consumption analysis, and 679 participants for iodine consumption analysis. This should be corrected.

Response: Thank You for your valuable observation. We made corrections to the manuscript: “. Salt and potassium intakes were estimated in 888 randomly selected participants by 24-hour urine sodium and potassium excretion and 679 individuals provided suitable 24-hour urine samples for the analysis of iodine excretion”.

Comment 3. The acronym UNICEF on Page 2 line 66 is incorrect. This should be corrected.

Response: Thank You for your comment. We made corrections to the manuscript. Although, UNICEF originally was called United Nations International Children’s Emergency Fund, now the organization is called United Nations Children’s Fund.

Comment 4. Figure 1 on Page 3 should be fixed. The numbers for the West and North regions are cut.

Response: Thank You for your comment. We made changes to the Figure 1 (added as a picture).

Comment 5. A supplemental document can be provided including the questionnaire given to the participants mentioned on Page 5 lines 144-153.

Response: Thank You for your comment, but we decided not to share excessive information which may withdraw attention from the main focus (consumption of salt, potassium and iodine).

Comment 6. Is the sentence on Page 5 line 163 another sub-section?

Response: Thank You for your comment. Yes, this is another sub-section. We made corrections.

Comment 7. The Materials and Methods (Page 5 line 167) mentions that chloride, albumin, and urea were determined in the urine. These results can be included in the manuscript to determine if participants suffer from kidney function and/or improper fluid cell balance.

Response: Thank You for your important opinion. We decided not to focus on kidney function in this article to keep focus on main aspects.  Therefore, we removed this information from the manuscript.

Comment 8. The percentage mentioned on Page 6 line 226 is for women. This should be corrected.

Response: Thank You for your comment. We made corrections.  

Comment 9. Is the measurement g/d or mmol/d the same as g/24h or mmol/24h? The manuscript should be consistent.

Response: Thank You for your comment. We changed all units/d to units/24h.

Comment 10. As a suggestion, Table 4 should be placed before Figure 3 on Page 8. In the manuscript Table 4 is mentioned first than Figure 3.

Response: Thank You for your comment. We received same comment from Reviewer 1. We made changes to the manuscript.

Comment 11. The p-value on Page 9 line 264 should be p<0.001.

Response: Thank You for your comment. We made changes to the manuscript.

Comment 12. The title of Iodine status should be µg/l, it's missing information.

Response: Thank You for your comment. We made changes to the manuscript.

Comment 13. The percentage on Page 10 line 295 should be 40.2%. In Figure 3 the number of participants that consumed 10-15 g of salt per day was 26% and 14.2% for participants that consumed > 15 g of salt per day.

Response: Thank You for your comment. We made changes to the manuscript.

Comment 14. Is the sentence on Page 12 line 412 another sub-section?

Response: Thank You for your comment. We made corrections and removed bullets.

Comment 15. The manuscript should focus on and mention that men have significantly different results leading to poorer outcomes than women. Where there any significant responses in the questionnaire that can answer why? Example: Demographic and Social status, Health status, or Lifestyle habits (physical activity).

Response: Thank You for your comment. We included additional information in the Discussion section: “Men usually have a higher food intake, which translates into a higher salt intake. In addition, Lithuanian men live unhealthy lifestyles. Our questionnaire revealed that women are less likely to eat salty processed meat, frozen and processed fish. Compared to women, a larger proportion of men consume crisps, roasted nuts, smoked, salted or cured meat products, prepared soups and stock cubes for infusion and drink natural mineral water. In addition, men use these products more often than women and eat food cooked outside the home more often. Moreover, the majority of men do not think about regulating their salt consumption”. 

Round 2

Reviewer 1 Report

The changes by the authors in response to the reviewers' comments have significantly improved the manuscript. This reviewer recommends publication.

Author Response

Thank You for reviewing the manuscript again!

Reviewer 2 Report

The previous comments on the manuscript have been addressed and the manuscript is clear to understand the purpose. However, minor comments are suggested:

1. What does NATRIJOD stand for on Page 3 line 102 and Figure 2?

2. There should be a semicolon (;) between "terminally or mentally ill and pregnant women" on Page 3 line 122.

3. Why does SRM3668 has an asterisk in Table 1?

4. Why do some p-values <0.001 have an asterisk in Table 2 and Table 3, and others that also have p-values <0.001 do not?

5. The paragraph on Page 8 lines 255-261 should include the values of the table, as it was previously written, so the reader doesn't have to go back and forth between the text and the table. The same comment for the paragraph on Page 10 lines 284-292, and on Page 11 line 318.

6. As a suggestion, Figure 3 could include the percentage values above the bars.

Author Response

Thank You for reviewing the manuscript again. Please find the response to the comments below:

Comment 1. What does NATRIJOD stand for on Page 3 line 102 and Figure 2?

Response: Thank You for Your observation. NATRIJOD – is the title of our study in Lithuanian. “Natri” is a short form of “Natris”, which means sodium and “JOD” is a short form of “Jodas” which means iodine in Lithuanian. But we decided to erase this title from article to avoid confusion.

Comment 2. There should be a semicolon (;) between "terminally or mentally ill and pregnant women" on Page 3 line 122.

Response: Thank You for Your comment. We made corrections.

Comment 3. Why does SRM3668 has an asterisk in Table 1?

Response: Thank You for Your remark. An asterisk was accidently left during adjustment process. We made corrections to the manuscript.

Comment 4. Why do some p-values <0.001 have an asterisk in Table 2 and Table 3, and others that also have p-values <0.001 do not?

Response: Thank You for Your comment. We marked p-values differently, calculated with different statistical methods. P-value indicates statistical difference between two groups (men and woman) and *P-value indicates the significance of the difference in proportion.

Comment 5. The paragraph on Page 8 lines 255-261 should include the values of the table, as it was previously written, so the reader doesn't have to go back and forth between the text and the table. The same comment for the paragraph on Page 10 lines 284-292, and on Page 11 line 318.

Response: Thank You for Your remark. We made corrections.

Comment 6. As a suggestion, Figure 3 could include the percentage values above the bars.

Response: Thank You for Your suggestion. We added percentage values.